# Compressive Sampling with Multiple Bit Spread Spectrum-Based Data Hiding

**Gelar Budiman** [1,2,*,†,‡](), **Andriyan Bayu Suksmono** [1,†] **and Donny Danudirdjo** [1,†]

[1]    School of Electrical Engineering and Informatics, Institut Teknologi Bandung, Bandung 40132, Indonesia; suksmono@stei.itb.ac.id (A.B.S.); donny@stei.itb.ac.id (D.D.)
[2]    School of Electrical Engineering, Telkom University, Bandung 40257, Indonesia
*    Correspondence: gelarbudiman@telkomuniversity.ac.id
†    Current address: Jl. Ganesha No.10, Bandung 40132, Indonesia.
‡    Current address: Jl. Telekomunikasi Terusan Buahbatu Bandung 40257, Indonesia.

**Abstract:** We propose a novel data hiding method in an audio host with a compressive sampling technique. An over-complete dictionary represents a group of watermarks. Each row of the dictionary is a Hadamard sequence representing multiple bits of the watermark. Then, the singular values of the segment-based host audio in a diagonal matrix are multiplied by the over-complete dictionary, producing a lower size matrix. At the same time, we embed the watermark into the compressed audio. In the detector, we detect the watermark and reconstruct the audio. This proposed method offers not only hiding the information, but also compressing the audio host. The application of the proposed method is broadcast monitoring and biomedical signal recording. We can mark and secure the signal content by hiding the watermark inside the signal while we compress the signal for memory efficiency. We evaluate the performance in terms of payload, compression ratio, audio quality, and watermark quality. The proposed method can hide the data imperceptibly, in the range of 729–5292 bps, with a compression ratio 1.47–4.84, and a perfectly detected watermark.

**Keywords:** compressive sampling; compressed sensing; watermark; data hiding; spread spectrum; singular value decomposition; Hadamard

## 1. Introduction

At present, the exchange of data and information in the Internet network has increased very dramatically. With more and more people accessing the Internet and more and more content that can be accessed, the size of the data accessed in a given time increases on an exponential scale. With the increase in data access, more and more crimes related to data include data falsification, data theft, claiming unilateral ownership of data, leaking data, deception of data, and many other crimes related to Internet data access. These problems have implications for the more losses experienced by data owners, which also affect state losses. Losses suffered by the state cause harm to its people, such that crime in the Internet world only benefits certain parties and results in a big loss for the wider community. Thus, technology that provides security for data, including marking ownership rights to data and hiding important data when sent over the Internet, becomes mandatory to anticipate losses suffered by the wider community.

With more and more data content accessed, the greater the memory capacity needed; besides, assuming the network infrastructure does not increase, the network capacity also decreases due to increased data traffic accessed, and power requirements on the network infrastructure also increase. These conditions bring problems in how to access data efficiently so that we can save the infrastructure and energy needs by minimum usage. One technique that can provide solutions to these problems is

Compressive Sampling or Compressed Sensing (CS). This technique takes or picks up part of the data or signal from the sensor and then sends the data from the sample, and the receiver can reconstruct it back to the data as if it were authentic.

In this paper, we propose a technique for sampling audio signals and inserting or hiding data into them at the same time, so that the sampled signals have a smaller size and, at the same time, there is data inserted into the encoded data. With this technique, the signal stored in the cloud system from recording results by sampling is smaller in size, and we can mark it with hidden data at the same time. The broadcast monitoring application is an example of how to monitor signals in real-time and store the results in the cloud. Monitoring such signals is more efficient if partial signal sampling is applied, such that the signal size becomes smaller than the original signal. At the same time, marking or indexing is applied by hiding data on the signal at any given duration to secure the authenticity of the monitored signal or to index the monitored signal by hiding its index on the encoded signal. Another application example is the recording of biomedical signals in which one samples them using several sensors and, at the same time, embeds the ownership marking or index into the encoded signal. Thus, the recorded biomedical signal has a smaller size than the original size, but does not reduce the quality of the biomedical signal, and there is a marking that is inserted in the encoded signal to secure the biomedical signal.

CS in audio combined with the data hiding technique is a rare topic. The combination of CS and data hiding makes it possible to compress the audio and, at the same time, hide the watermark. Hua in [1] and Xin in [2] formerly proposed the CS applications in audio combined with data hiding. In [2], Xin proposed an embedding method on host audio that was semi-fragile zero-watermarking by decomposing the audio into the wavelet domain and applying the CS technique to the audio wavelet coefficients without describing the audio reconstruction to determine the audio quality after the embedding process. Watermarks inserted in the measurement vector utilize positive and negative signs on the matrix elements. The result is that the inserted watermark is resistant to damaging samples from the signal. However, this paper does not explain the function of CS in terms of reducing the signal size. Xin only explained CS techniques as a technique of inserting data with the property of being semi-fragile.

Griffin, in [3], proposed the CS method to compress the sinusoidal signal. Griffin investigated whether CS can be used to compress sinusoidal audio at a low speed because audio models like this have a high degree of spacing in the frequency domain. In his proposed method, Griffin performed CS techniques on single channels and multiple channels of audio signals with sinusoidal characteristics only. Griffin stated that the research he did was not to develop audio compression techniques and compare with existing compression techniques, but to find out how far CS was able to be applied in reducing the size of audio files so that the application applied, in this case, was for wireless sensor networks. Griffin could produce the smallest compression ratio by 5.4%. He applied spectral whitening first on the new audio, then applied the CS technique to the spectral results, so that he produced a tiny compression ratio with a good quality of the reconstruction results.

Fakhr in [4] proposed an insertion method using CS techniques by first thinning the host audio and watermark signals using the Walsh-Hadamard Transform (WHT), Discrete Cosine Transform (DCT), and Karhunen–Loeve Transform (TLC). Watermark extraction and the audio host are done by reconstruction $L_1$ minimization. Fakhr claimed that the technique could withstand MP3 attacks at the lowest rate of 64 kbps with an 11 bps watermark payload and the highest payload at 172 bps against additive noise attacks. However, Fakhr used CS not for compression techniques, but as an insert technique. Fakhr used MP3 attacks as compression to reduce the size of the audio signal after embedding a watermark. The watermarking applied to the compressed sensing domain was also proposed by Jeng-Shyang in [5,6]. Jeng-Shyang used DWT-DCT as the host sparsity before he applied the CS acquisition and the watermark embedding procedure. In the other scenario, Patsakis in [7] used CS to detect the embedded data by Least Significant Bit (LSB) and the DCT method. CS in [7]

was used as a denoising filter to detect the hidden data. However, the embedding method in [5–7] worked on the image as the host.

In [1], Hua proposed a data hiding technique that was combined with CS synthetically. Suppose we define an over-complete dictionary $\mathbf{A} \in \mathbb{R}^{p \times r}$, an uncompressed vector $\mathbf{z} \in \mathbb{R}^{r \times 1}$, a watermark bit to be inserted as $b \in \{-1, +1\}$, a watermark code sequence $\mathbf{w} \in \mathbb{R}^{r \times 1}$, a compressed vector $\mathbf{y} \in \mathbb{R}^{p \times 1}$, and $\alpha$ as the gain control of the watermark, then we have:

$$\mathbf{y} = \mathbf{A}(\mathbf{z} + \alpha b \mathbf{w}), \tag{1}$$

Hua inserted $b$ as the additional operation to $\mathbf{z}$ after multiplying by $\alpha \mathbf{w}$. In this paper, we embed the watermark bits into the over-complete matrix $\mathbf{A}$. Then, we multiply $\mathbf{A}$ by the diagonal matrix from the singular values of the host audio.

The data hiding technique proposed in this paper is multiple orthogonal codes based on Spread Spectrum (SS), as formerly introduced by Xin in [8] in time domain embedding and continued by Xiang in DCT domain embedding in [9,10]. We use the Hadamard code as the sequence for multiple bits of the watermark due to its best code performance [11]. The matrix $\mathbf{A}$ consists of $p$ Hadamard sequences that represent $p$ groups of multiple bits.

One of the signal sparsity techniques is a shrinkage technique on Singular Value Decomposition (SVD) output. This SVD technique truncates $\mathbf{U}$, $\mathbf{S}$, and $\mathbf{V}$ with a specific rank as also described in [12–15]. This shrinkage technique yields a more compressed signal as the CS output, but certainly decreases the quality of the reconstructed signal. In this paper, we decompose a host signal using SVD. Then, the outputs of SVD, i.e., $\mathbf{U}$, $\mathbf{S}$, and $\mathbf{V}$, are truncated at a specific rank. We transform the truncated singular matrix $\mathbf{S}_r$ to compressed domain $\mathbf{Y}$ via an over-complete dictionary containing SS-based data hiding $\mathbf{A}$. Thus, the matrices ready to be transmitted to the detector are $\mathbf{U}_r$, $\mathbf{Y}$, and $\mathbf{V}_r$. Then, in the receiver, firstly, we detect dictionary $\mathbf{A}$ containing the hidden data. We can extract the hidden data from the detected dictionary. Not only can we take back the hidden data, but also, we can get the reconstructed signal to the original domain. Note that the process on the receiver needs only the compressed domain signal, such as $\mathbf{U}_r$, $\mathbf{Y}$, and $\mathbf{V}_r$. There is no dictionary, and original data are needed for data detection and signal reconstruction.

We organize the rest of this paper as follows. Section 2 describes the sparsity of the singular value and CS technique for the audio compression. Section 3 explains the mathematics model and derivation of audio watermarking including the embedding, the extraction, the audio reconstruction process, and the effect of the noisy environment in this proposed method. Section 4 discusses the result of the simulation, while Section 5 reports the conclusion of this paper.

## 2. Sparse Singular Value and CS Technique

The host signal in the form of a vector $\mathbf{x} = [x_1, x_2, \cdots x_L] \in \mathbb{R}^{1 \times L}$ is converted into a two-dimensional matrix $\mathbf{X} \in \mathbb{R}^{M \times M}$ where $L = M^2$. The conversion to a two-dimensional matrix $X$ is applied in such a way that it produces:

$$\mathbf{X} = \begin{bmatrix} x_1 & x_{M+1} & \cdots & x_{M(M-1)+1} \\ x_2 & x_{M+2} & \cdots & x_{M(M-1)+2} \\ \vdots & \vdots & \ddots & \vdots \\ x_M & x_{2M} & \cdots & x_{M^2} \end{bmatrix}. \tag{2}$$

The SVD process of $\mathbf{X}$ obtains orthogonal matrices $\mathbf{U} \in \mathbb{R}^{M \times M}$, $\mathbf{S} \in \mathbb{R}^{M \times M}$, and $\mathbf{V} \in \mathbb{R}^{M \times M}$, where the relationship is described as:

$$\mathbf{X} = \mathbf{U}\mathbf{S}\mathbf{V}^T, \tag{3}$$

where $\mathbf{S}$ is a sparse diagonal matrix having $M$ non-zero elements in the diagonal of the matrix as $M$ singular values. For compression needs, $\mathbf{U}$, $\mathbf{S}$, and $\mathbf{V}$ can be truncated or reduced to $\mathbf{U_r} =$

$\mathbf{U}[1, .., M; 1, .., r] \in \mathbb{R}^{M \times r}$, $\mathbf{S_r} = \mathbf{S}[1, .., r; 1, .., r] \in \mathbb{R}^{r \times r}$, and $\mathbf{V_r} = \mathbf{V}[1, .., M; 1, .., r] \in \mathbb{R}^{M \times r}$ with $r < M$. Then, we apply CS acquisition $\mathbf{S_r}$ as:

$$\mathbf{Y} = \mathbf{A}\mathbf{S_r}, \tag{4}$$

where $\mathbf{A} \in \mathbb{R}^{p \times r}$ is an over-complete dictionary containing the SS-based encoded watermark and $\mathbf{Y} \in \mathbb{R}^{p \times r}$ is an output of CS acquisition with a smaller size than $\mathbf{S}$. The truncated matrix $\mathbf{S_r}$ has the form of:

$$\mathbf{S_r} = \begin{bmatrix} \sigma_1 & 0 & \cdots & 0 \\ 0 & \sigma_2 & \cdots & 0 \\ \vdots & \vdots & \ddots & \vdots \\ 0 & 0 & \cdots & \sigma_r \end{bmatrix}, \tag{5}$$

where $\sigma_1, \sigma_2, ..., \sigma_r$ are the singular value elements. The matrix $\mathbf{A}$ is described later in Section 3.1. Finally, we have three matrices to be transmitted, that is $\mathbf{U_r}$, $\mathbf{V_r}$, and $\mathbf{Y}$. From this result, we can calculate the Compression Ratio (CR) as the comparison between the original signal length and the transmitted signal length as:

$$\text{CR} = \frac{L_X}{L_T} = \frac{M^2}{2Mr + pr}, \tag{6}$$

where $L_X$ is the element number of $\mathbf{X}$, that is $M^2$, and $L_T$ is the total number of the transmitted elements $\mathbf{U_r}$, $\mathbf{Y}$, and $\mathbf{V_r}$, i.e., $2Mr + pr$.

We can calculate the reconstructed audio matrix with the same size as $\mathbf{X}$ in the form of:

$$\mathbf{X_r} = \mathbf{U_r}\widehat{\mathbf{S}}_r{\mathbf{V_r}}^T = \begin{bmatrix} \hat{x}_1 & \hat{x}_{M+1} & \cdots & \hat{x}_{M(M-1)+1} \\ \hat{x}_2 & \hat{x}_{M+2} & \cdots & \hat{x}_{M(M-1)+2} \\ \vdots & \vdots & \ddots & \vdots \\ \hat{x}_M & \hat{x}_{2M} & \cdots & \hat{x}_{M^2} \end{bmatrix}, \tag{7}$$

where $\mathbf{X_r} \in \mathbb{R}^{M \times M}$, but its element values are slightly different than $\mathbf{X}$. The $r$ value controls the signal quality and the signal compression ratio. If $r$ is lower, then the compression ratio is higher, but the signal quality is worse. Finally, we can get $\hat{\mathbf{x}} = [\hat{x}_1, \hat{x}_2, \cdots \hat{x}_{M^2}]$ as a reconstructed or decompressed version of the signal by converting two-dimensional matrix $\mathbf{X_r}$ back to a vector or one-dimensional signal $\hat{\mathbf{x}}$; thus, we can calculate the signal quality by comparing $\mathbf{x}$ and $\hat{\mathbf{x}}$.

## 3. Data Hiding Model

### 3.1. An Overcomplete Dictionary with SS-Based Content

In this proposed method, firstly, we convert the audio host to the frequency domain using DCT in the process before applying insertion and compression. In the audio receiver, after being reconstructed or decompressed, the reconstructed audio is re-converted to the time domain with IDCT. The DCT and Inverse DCT (IDCT) formulations used for this method are in the following equation [16]:

$$X(k) = w(k) \sum_{n=0}^{N_p-1} x(n) \cos\left(\frac{\pi(2n-1)(k-1)}{2N_p}\right) \tag{8}$$

$$x(n) = \frac{2}{N_p+1} \sum_{k=0}^{N_p-1} l(k)X(k) \cos\left(\frac{\pi(2n-1)(k-1)}{2N_p}\right), \tag{9}$$

where $X(k)$ is the audio signal in the DCT domain, $x(n)$ is the audio signal in the time domain, and $N_p$ is the number of DCT points. $l(k)$ is defined in the following equation:

$$l(k) = \begin{cases} \frac{1}{\sqrt{N}}, & \text{if } k = 0 \\ \frac{2}{\sqrt{N}}, & \text{if } 1 \le k \le N_p - 1 \end{cases}. \tag{10}$$

After transforming the signal to the frequency domain by DCT, we apply the signal to the SVD decomposition as displayed in Figure 1. In detail, compression and embedding procedure is described at Table 1.

In this paper, the orthogonal code mapping to multiple bit watermarks is a Hadamard sequence taken from the Hadamard matrix. Denote the Hadamard matrix $\mathbf{H}_r \in \{-1, +1\}^{r \times r}$ generated by [17,18] as:

$$\mathbf{H}_r = \begin{bmatrix} \mathbf{H}_{r\backslash 2} & \mathbf{H}_{r\backslash 2} \\ \mathbf{H}_{r\backslash 2} & -\mathbf{H}_{r\backslash 2} \end{bmatrix}, \tag{11}$$

where $\mathbf{H}_1 = [1]$. Assume $\mathbf{H}_r(j)$ is a vector from the $j^{\text{th}}$ row of $\mathbf{H}_r$, then the orthogonal Hadamard sequence $\mathbf{p}_j$, where $j = 1, 2, ..., r$, is obtained from:

$$\mathbf{p}_j = \mathbf{H}_r(j). \tag{12}$$

Let $\mathbf{A}_0 \in \{-1, +1\}^{p \times r}$ be an SS-based content matrix, where $p < r$, and $\mathbf{p}_{t_i} \in \mathbb{R}^{1 \times r}$ is a Hadamard sequence associated with the watermark bits in the $i^{\text{th}}$ row of $\mathbf{A}_0$. Let $t_i = \{t_1, t_2, ..., t_p\}$ be the set of Hadamard sequence indices where $i$ is a row index of $\mathbf{A}_0$. Thus, $\mathbf{A}_0$ contains $\mathbf{p}_{t_i}$ as:

$$\mathbf{A}_0 = \begin{bmatrix} \mathbf{p}_{t_1} \\ \mathbf{p}_{t_2} \\ \vdots \\ \mathbf{p}_{t_p} \end{bmatrix}, \tag{13}$$

where the semicolon from (13) restricts each $\mathbf{p}_{t_i}$ to the different row. Since there are $p$ rows of $\mathbf{A}_0$, there are $p$ Hadamard sequences in $\mathbf{A}_0$. Thus, we have an over-complete dictionary $\mathbf{A} \in \mathbb{R}^{p \times r}$:

$$\mathbf{A} = \frac{1}{p} \mathbf{A}_0, \tag{14}$$

with the unit norm of its columns: $\|a_m\|_2^2 = 1$, where $m = 1, 2, ..., r$.

A Hadamard sequence represents multiple watermark bits. Assume that there are $N_s$ watermark bits for a Hadamard sequence, then there are $N_p$ different Hadamard sequence possibilities, where $N_p = 2^{N_s}$. Note that the length of a Hadamard sequence and also the row of matrix $\mathbf{A}$ is $r$ bits, thus $r = N_p$ due to the square size of Hadamard matrix as (11). Denote $\mathbf{w}_{t_i}$ as a watermark vector in the $i^{\text{th}}$ segment of the watermark with a vector index or a Hadamard index $t_i$, then:

$$\mathbf{w}_{t_i} = [w_{t_i}(1) \quad w_{t_i}(2) \quad \cdots \quad w_{t_i}(N_s)], \tag{15}$$

where $w_{t_i}(l) \in \{-1, +1\}$ and $l = 1, 2, ..., N_s$. In multi-bit SS, the watermark vector $\mathbf{w}_{t_i}$ is mapped to a Hadamard sequence $\mathbf{p}_{t_i}$. For example, if we have three bits watermarked in a Hadamard sequence, or $N_s = 3$ bits, then $N_p = 2^{N_s} = 8$ bits; thus, all watermark possibilities and their mapping to Hadamard sequences are displayed in Table 2. If we have two segments or two vectors of watermark $\mathbf{w}_{t_1} = [-1, +1, -1]$ and $\mathbf{w}_{t_2} = [+1, -1, -1]$, then using Table 2, we get $t_1 = 3$ and $t_2 = 5$; thus, $\mathbf{p}_{t_1} = \mathbf{p}_3 = \{+1, +1, -1, -1, +1, +1, -1, -1\}$, and $\mathbf{p}_{t_2} = \mathbf{p}_5 = \{+1, +1, +1, +1, -1, -1, -1, -1\}$.

**Table 1.** Embedding process.

| | |
|---|---|
| Step 1: | Read a host signal $x(n)$, and transform it into the frequency domain by DCT $L$-point obtaining $X(k)$ |
| Step 2: | Reshape $X(k)$ in $L$ and sample it to a 2D square matrix producing $\mathbf{X}$ with size $M \times M$ |
| Step 3: | Decompose $\mathbf{X}$ to $\mathbf{U}$, $\mathbf{S}$, and $\mathbf{V}$ using SVD |
| Step 4: | Reduce the matrix size of $\mathbf{U}$, $\mathbf{S}$, and $\mathbf{V}$ with rank $r$ to $\mathbf{U_r}$, $\mathbf{S_r}$, and $\mathbf{V_r}$ |
| Step 5: | Generate the $\mathbf{A}$ matrix containing $p$ Hadamard sequences by mapping each multi-watermark bit to an associated random Hadamard sequence using (13) |
| Step 6: | Apply CS acquisition to $\mathbf{A}$ and $\mathbf{S_r}$ by (4), producing $\mathbf{Y}$ |
| Step 7: | Transmit the compressed signal with hidden data represented using $\mathbf{U_r}$, $\mathbf{Y}$, and $\mathbf{V_r}$ |

**Table 2.** Watermarks and Hadamard sequences' example for $N_s = 3$, $N_p = 8$, and $r = 8$.

| Index ($t_i$) | Watermark Bits ($w_i$) | Hadamard Sequence ($\mathbf{p}_{t_i}$) |
|---|---|---|
| 1 | {−1,−1,−1} | {+1,+1,+1,+1,+1,+1,+1,+1} |
| 2 | {−1,−1,+1} | {+1,−1,+1,−1,+1,−1,+1,−1} |
| 3 | {−1,+1,−1} | {+1,+1,−1,−1,+1,+1,−1,−1} |
| 4 | {−1,+1,+1} | {+1,−1,−1,+1,+1,−1,−1,+1} |
| 5 | {+1,−1,−1} | {+1,+1,+1,+1,−1,−1,−1,−1} |
| 6 | {+1,−1,+1} | {+1,−1,+1,−1,−1,+1,−1,+1} |
| 7 | {+1,+1,−1} | {+1,+1,−1,−1,−1,−1,+1,+1} |
| 8 | {+1,+1,+1} | {+1,−1,−1,+1,−1,+1,+1,−1} |

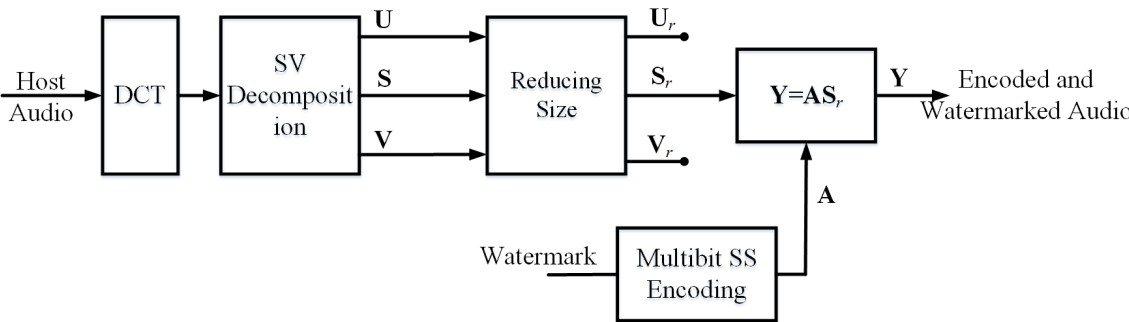

**Figure 1.** Watermark embedding and audio encoding.

The over-complete matrix $\mathbf{A}_0$ contains $pN_s$ bits of watermark for the host with length $M^2$; thus, we can compute watermark payload $C$ in bps as:

$$C = \frac{pN_s F_s}{M^2}, \tag{16}$$

where $F_s$ is the host signal sampling rate in samples/s. Due to $N_s = \lceil \log_2 N_p \rceil = \lceil \log_2 r \rceil$, thus (16) will be as:

$$C = \frac{p\lceil \log_2 r \rceil F_s}{M^2}. \tag{17}$$

Once $\mathbf{A}$ is generated from the associated watermark bits, it is embedded into $\mathbf{S_r}$ using matrix multiplication in (4). The result $\mathbf{Y}$ is not only a matrix with a smaller size than $\mathbf{S_r}$, but also, it is embedded by the watermark bits. The matrix $\mathbf{S_r}$ is a diagonal matrix whose size is reduced from the original one $\mathbf{S}$. From (4), (5), and (13), the equation $\mathbf{Y} = \mathbf{AS_r}$ can be exploited as:

$$\begin{bmatrix} \mathbf{y}_{t_1} \\ \mathbf{y}_{t_2} \\ \vdots \\ \mathbf{y}_{t_p} \end{bmatrix} = \begin{bmatrix} \mathbf{p}_{t_1} \\ \mathbf{p}_{t_2} \\ \vdots \\ \mathbf{p}_{t_p} \end{bmatrix} \begin{bmatrix} \sigma_1 & 0 & \cdots & 0 \\ 0 & \sigma_2 & \cdots & 0 \\ \vdots & \vdots & \ddots & \vdots \\ 0 & 0 & \cdots & \sigma_r \end{bmatrix}, \tag{18}$$

where $\mathbf{y}_{t_i} \in \mathbb{R}^{1 \times r}$ is a vector of matrix $\mathbf{Y}$ at row $i$, which also corresponds to $\mathbf{p}_{t_i}$, and $\sigma_1, \sigma_2, ..., \sigma_r$ are singular value elements of $\mathbf{S_r}$. Each row of $\mathbf{A}$ or $\mathbf{p}_{t_i}$ is a vector with size $1 \times r$. $\mathbf{S_r}$ is a diagonal matrix with size $r \times r$. Thus, we can simplify (18) to the following equation:

$$\begin{bmatrix} \mathbf{y}_{t_1} \\ \mathbf{y}_{t_2} \\ \vdots \\ \mathbf{y}_{t_p} \end{bmatrix} = \begin{bmatrix} \mathbf{p}_{t_1}\mathbf{S_r} \\ \mathbf{p}_{t_2}\mathbf{S_r} \\ \vdots \\ \mathbf{p}_{t_p}\mathbf{S_r} \end{bmatrix} \tag{19}$$

Then, we can have the following simple vector expression:

$$\mathbf{y}_{t_i} = \mathbf{p}_{t_i}\mathbf{S_r}. \tag{20}$$

### 3.2. Data and Dictionary Detection

Once we get the compressed and watermarked signal $\mathbf{Y}$ or $\mathbf{y}_i$, it is transmitted to the receiver; thus, we get the received signal $\mathbf{Y}'$ or $\mathbf{y}'_i$. The received signals along $\mathbf{y}'_i$ are $\mathbf{U}'_\mathbf{r}$ and $\mathbf{V}'_\mathbf{r}$, as described in Section 2. One can choose whether to decompress the signal or to extract the watermark. Anyway, to decompress the signal, we need $\mathbf{A}$ or $\mathbf{p}_{t_i}$ using (22) for reconstructing $\mathbf{y}'_i$ to get $\widehat{\mathbf{S}}_\mathbf{r}$. It is clear that, either to extract the watermark or to decompress the signal, extracting $\mathbf{A}$ from $\mathbf{y}'_i$ is the first thing to be applied in the receiver since the compression and data hiding process is blind. Once we get $\mathbf{A}$, then we can extract the data, or we can reconstruct $\mathbf{y}'_i$ with detected $\mathbf{A}$ to obtain $\widehat{\mathbf{S}}'_\mathbf{r}$ using (37), (39), (40), and (41). Thus, we can use SVD reconstruction for $\widehat{\mathbf{S}}'_\mathbf{r}$, $\mathbf{U}'_\mathbf{r}$, and $\mathbf{V}'_\mathbf{r}$ to obtain a square matrix or $\mathbf{X}'_\mathbf{r}$ using (7). Finally, we get the reconstructed signal $\mathbf{x}'$ by converting the two-dimensional matrix $\mathbf{X}'_\mathbf{r}$ to the vector $\mathbf{x}'$. Clearly, the detection and reconstruction procedure is displayed in Figure 2 and Table 3.

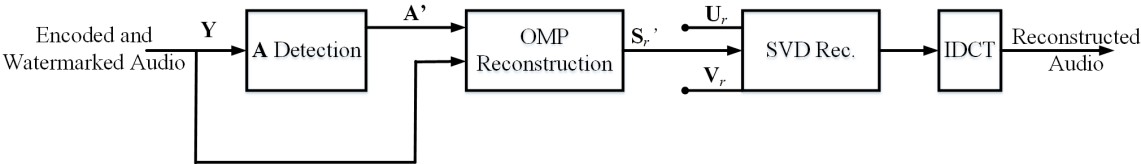

**Figure 2.** Watermark detection and audio decoding.

**Table 3.** Detection and reconstruction process.

| | |
|---|---|
| Step 1: | Detect $t_i$ from $\mathbf{Y}'$ using (22) for extracting the hidden data |
| Step 2: | Associate detected $t_i$ with $\mathbf{p}_{t_i}$, and form $\widehat{\mathbf{A}}$ using (13) |
| Step 3: | Reconstruct $\mathbf{Y}'$ using $\widehat{\mathbf{A}}$ by (37), (39), (40), and (41) to obtain $\mathbf{S}'_\mathbf{r}$ |
| Step 4: | Reconstruct $\mathbf{U}_\mathbf{r}$, $\mathbf{S}'_\mathbf{r}$, and $\mathbf{V}_\mathbf{r}$ by SVD reconstruction to obtain the decompressed signal in 2D matrix $\mathbf{X}'_\mathbf{r}$ by (7) |
| Step 5: | Reshape 2D matrix $\mathbf{X}'_\mathbf{r}$ to a 1D matrix, obtaining $X'(k)$ |
| Step 6: | Transform $X'(k)$ to the time domain by the IDCT $L$-point, obtaining the reconstructed signal $x'(n)$ |

For $\mathbf{p}_{t_i}$ detection, we need to correlate $\mathbf{y}'_{t_i}$ to $\mathbf{p}^T_j$ as:

$$K_{ij} = \left| \mathbf{y}'_{t_i}\mathbf{p}^T_j \right|, \tag{21}$$

where $i = 1, 2, .., p$ and $j = 1, 2, .., N_p$. From (21), there is an index of $j$ whose correlation $K_{ij}$ is the highest, that is $j = t$. Thus, the formula to detect the correct index of the Hadamard sequence embedded into $\mathbf{y}'_{t_i}$ is:

$$t_i = \underset{j \in \{1,2,...,N_p\}}{\mathrm{argmax}} \left| \mathbf{y}'_{t_i}\mathbf{p}^T_j \right|. \tag{22}$$

Since we can detect $t_i$, we decode the detected Hadamard code to the associated watermark bits according to one-to-one mapping between the index, the Hadamard code, and the associated watermark bits. For detection proving needs, assume there is no attack, then $\mathbf{y}'_{t_i} = \mathbf{y}_{t_i}$. Thus, (21) is:

$$K_{ij} = \left| \mathbf{y}_{t_i} \mathbf{p}_j^T \right|. \tag{23}$$

Substituting (20) into (23) results in:

$$K_{ij} = \left| \mathbf{p}_{t_i} \mathbf{S_r} \mathbf{p}_j^T \right|. \tag{24}$$

Assume that $t_i = j$; thus, $\mathbf{p}_{t_i} = \mathbf{p}_j$, then (24) is an autocorrelation as:

$$K_a = \left| \mathbf{p}_j \mathbf{S_r} \mathbf{p}_j^T \right|. \tag{25}$$

Assume that $\mathbf{p}_j$ consists of such elements as:

$$\mathbf{p}_j = \begin{bmatrix} p_{j_1} & p_{j_2} & \cdots & p_{j_r} \end{bmatrix}; \tag{26}$$

therefore, (24) becomes:

$$K_a = \left| \begin{bmatrix} p_{j_1} \\ p_{j_2} \\ \vdots \\ p_{j_r} \end{bmatrix}^T \begin{bmatrix} \sigma_1 & 0 & \cdots & 0 \\ 0 & \sigma_2 & \cdots & 0 \\ \vdots & \vdots & \ddots & \vdots \\ 0 & 0 & \cdots & \sigma_r \end{bmatrix} \begin{bmatrix} p_{j_1} \\ p_{j_2} \\ \vdots \\ p_{j_r} \end{bmatrix} \right|. \tag{27}$$

By a matrix multiplication operation, (27) is described as:

$$K_a = \left| p_{j_1}^2 \sigma_1 + p_{j_2}^2 \sigma_2 + \cdots + p_{j_r}^2 \sigma_r \right| = \left| \sum_{i=1}^{r} p_{j_i}^2 \sigma_i \right|. \tag{28}$$

Since $\sigma_i > 0$ and $p_{j_i}^2 \gg 0$ for all $j$ and all $i$, then (28) becomes:

$$K_a = \sum_{i=1}^{r} p_{j_i}^2 \sigma_i \gg 0. \tag{29}$$

If $\mathbf{p}_{t_i} = \mathbf{p}_k$ and $\mathbf{p}_{t_i} \neq \mathbf{p}_j$, then (24) is a cross-correlation as:

$$K_c = \left| \mathbf{p}_k \mathbf{S_r} \mathbf{p}_j^T \right|$$

$$= \left| \begin{bmatrix} p_{k_1} \\ p_{k_2} \\ \vdots \\ p_{k_r} \end{bmatrix}^T \begin{bmatrix} \sigma_1 & 0 & \cdots & 0 \\ 0 & \sigma_2 & \cdots & 0 \\ \vdots & \vdots & \ddots & \vdots \\ 0 & 0 & \cdots & \sigma_r \end{bmatrix} \begin{bmatrix} p_{j_1} \\ p_{j_2} \\ \vdots \\ p_{j_r} \end{bmatrix} \right| \tag{30}$$

$$= \left| p_{k_1} p_{j_1} \sigma_1 + p_{k_2} p_{j_2} \sigma_2 + \cdots + p_{k_r} p_{j_r} \sigma_r \right|$$

$$= \left| \sum_{i=1}^{r} p_{k_i} p_{j_i} \sigma_i \right|. \tag{31}$$

Since $\mathbf{p}_k$ is mutually orthogonal with $\mathbf{p}_j$, it is confirmed that $K_a$ is comparable to $K_c$ with the following inequality:

$$K_a \gg K_c, \tag{32}$$

which means that the autocorrelation of the same Hadamard sequence is still much higher than the cross-correlation of the different Hadamard sequence on the singular value intervention. It confirms that the Hadamard sequence can be detected successfully; thus, from (22), $t_i$ is detected for $t_i = \{t_1, t_2, ..., t_p\}$, then we can obtain the associated watermark bits $\hat{\mathbf{w}}_{t_i} = \{\hat{\mathbf{w}}_{t_1}, \hat{\mathbf{w}}_{t_2}, ..., \hat{\mathbf{w}}_{t_p}\}$ and also all Hadamard sequences $\hat{\mathbf{p}}_{t_i} = \{\hat{\mathbf{p}}_{t_1}, \hat{\mathbf{p}}_{t_2}, ...\hat{\mathbf{p}}_{t_p}\}$, which form $\widehat{\mathbf{A}}$ using (13) and (14) as:

$$\widehat{\mathbf{A}} = \frac{1}{p} \begin{bmatrix} \hat{\mathbf{p}}_{t_1} \\ \hat{\mathbf{p}}_{t_2} \\ \vdots \\ \hat{\mathbf{p}}_{t_p} \end{bmatrix}, \tag{33}$$

where $p$ is the row number of $\widehat{\mathbf{A}}$. This procedure assures that there is no dictionary needed to detect the hidden data and also to reconstruct the signal, since the associated watermark bits $\hat{\mathbf{w}}_{t_i}$ are detected. Thus we can calculate the Bit Error Rate (BER) as a robustness parameter. The following equation is a BER formula:

$$\text{BER} = \frac{\sum_{i=1}^{L_w} |\hat{w}_i - w_i|}{L_w}, \tag{34}$$

where $w_i$ is the original watermark bit, $\hat{w}_i$ is the detected watermark bit, and $L_w$ is the total number of watermark bits.

### 3.3. Security Model

The Hadamard matrix is easily generated as described in (11). Anyone can attempt with the Hadamard matrix to reconstruct the dictionary to detect the hidden data and also to reconstruct the audio. This leads to insecure watermark bits hidden in the host audio, and accordingly, we apply a procedure to secure the Hadamard matrix as also discussed in [19–21]. The Hadamard matrix is multiplied by $-1$ at the row and the column of the matrix in a random manner. Denote $l_i \in \{1, r\}$ as an integer random permutation value where $i = 1, 2, ..., N_l$, and $N_l$ is the number of the generated integer random permutation value. Denote $\mathbf{H}_s$ as a secured Hadamard matrix, $\mathbf{H}_s(j)$ as a vector from the $j^{\text{th}}$ row of $\mathbf{H}_s$, and $\mathbf{H}_s^T(j)$ as a vector from the $j^{\text{th}}$ column of $\mathbf{H}_s$, then the security model of the Hadamard matrix after initial definition $\mathbf{H}_s = \mathbf{H}_r$ is defined as:

$$\begin{aligned} \mathbf{H}_s(l_i) &= -\mathbf{H}_r(l_i) \\ \mathbf{H}_s^T(l_i) &= -\mathbf{H}_r^T(l_i) \end{aligned} \tag{35}$$

The above procedure is repeated $N_l$ times from $l_1$ to $l_{N_l}$. Thus, with the secured Hadamard matrix, (12) is replaced by:

$$\mathbf{p}_j = \mathbf{H}_s(j). \tag{36}$$

Note that $\mathbf{H}_s$ is not only needed in the embedding process, but also in the detection/extraction process. However, it is not needed to pass $\mathbf{H}_s$ to the detector directly. We only pass $l_i$ as the integer random permutation value to the detector as the security key. By the procedure (35), $\mathbf{H}_s$ can be generated in the detector using $l_i$ as the key. According to [19,20], the modified Hadamard matrix combination using (35) has $(r!2^r)^2$ possibilities. For example, if $r = 16$, the number of modified Hadamard matrix is $1.88 \times 10^{36}$ possibilities. If the simulation needs one second to run the detection and reconstruction process using one Hadamard matrix, then it needs $1.88 \times 10^{36}$ seconds or $5.962 \times 10^{28}$ years using all Hadamard matrix possibilities. This confirms that this proposed security model is appropriate and meets the security requirement for the embedding and compression process.

### 3.4. Signal Reconstruction

Once $\widehat{\mathbf{A}}$ is obtained, $\widehat{\mathbf{S}}_{\mathbf{r}}$ reconstruction is simply solved by Orthogonal Matching Pursuit (OMP) [22,23]. The reconstruction process is carried out on each column of $\mathbf{Y}$ in sequence with $\widehat{\mathbf{A}}$ as a dictionary. Let $\mathbf{y}_m$ as a vector taken from the $m^{\text{th}}$ column of $\mathbf{Y}$, then for a general case, we can find the row position of the strongest atom as:

$$q_m = \underset{i \in \{1,2,\dots,p\}}{\text{argmax}} \ \widehat{\mathbf{A}}^T \mathbf{y}_m. \tag{37}$$

For a specific case, i.e., a singular matrix solution as the reconstructed one, the position of the highest atoms are indeed known, then (37) can be simplified as:

$$q_m = m. \tag{38}$$

Denote $\mathbf{a}_r$ as a vector taken from the $r^{\text{th}}$ column of $\widehat{\mathbf{A}}$, then we take a column of $\widehat{\mathbf{A}}$, which makes the strongest atom as:

$$\nabla = \mathbf{a}_{q_m}. \tag{39}$$

We reconstruct a non-zero element of $\widehat{\mathbf{S}}_{\mathbf{r}}$ in column $m$ by:

$$s_{q_m} = \left( \nabla^T \nabla \right)^{-1} \nabla^T \mathbf{y}_m. \tag{40}$$

This reconstruction procedure including (37), (39), and (40) is repeated $r$ times with the increment of $m$, thus obtaining:

$$\widehat{\mathbf{S}}_{\mathbf{r}} = \begin{bmatrix} s_{q_1} & 0 & \cdots & 0 \\ 0 & s_{q_2} & \cdots & 0 \\ \vdots & \vdots & \ddots & \vdots \\ 0 & 0 & \cdots & s_{q_r} \end{bmatrix}. \tag{41}$$

Then, the next step is to form the signal by SVD reconstruction, as described in (7). Thus, finally, we can compute the signal quality.

### 3.5. Noisy Environment

Note that the compressed and watermarked audio in this paper is the coded audio. A human cannot directly listen to the coded audio without decoding it first. This means that the signal processing attacks against the coded audio are not the same as the attacks against the real audio signal. The signal processing attacks against the real audio signal were standardized in the Stirmark benchmark [24]. However, the Stirmark benchmark is not appropriate for the robustness evaluation of this proposed method except for the additive noise attack. The additive noise attack is the signal processing attack that we can generally use to evaluate the watermarking compression robustness. In the real situation, this additive noise attack in the receiver happens due to the existing thermal condition of the hardware. In this subsection, we describe mathematically how our proposed method is robust to additive noise attack. If the compressed and watermarked signal $\mathbf{y}_i$ is under an additive noise environment, then (23) becomes:

$$\begin{aligned} K_{ij} &= \left| (\mathbf{y}_i + \mathbf{n}_i) \, \mathbf{p}_j^T \right| \\ &= \left| (\mathbf{p}_{t_i} \mathbf{S}_{\mathbf{r}} + \mathbf{n}_i) \, \mathbf{p}_j^T \right| . \\ &= \left| \mathbf{p}_{t_i} \mathbf{S}_{\mathbf{r}} \mathbf{p}_j^T + \mathbf{n}_i \mathbf{p}_j^T \right| \end{aligned} \tag{42}$$

Assume $\mathbf{p}_{t_i} = \mathbf{p}_j$, then (42) becomes:

$$K_{ij} = \left| \mathbf{p}_j \mathbf{S}_{\mathbf{r}} \mathbf{p}_j^T + \mathbf{n}_i \mathbf{p}_j^T \right| . \tag{43}$$

Because $\mathbf{n}_i$ is independent of $\mathbf{p}_j^T$, thus $\mathbf{p}_j\mathbf{S_r}\mathbf{p}_j^T \gg \mathbf{n}_i\mathbf{p}_j^T$, then (43) becomes:

$$K_{ij} \approx \left| \mathbf{p}_j\mathbf{S_r}\mathbf{p}_j^T \right| = K_a. \tag{44}$$

Thus, we confirm that the data inserted with the proposed method can be detected even in the additive noise environment. The performance evaluation of the proposed method, when attacked by additive noise, depends on the power ratio between the host audio and the additive noise represented by the Signal-to-Noise power Ratio (SNR) with the following formula:

$$\text{SNR} = 10\log_{10}\left( \frac{\sum\limits_{i=1}^{r} y_i^2}{\sum\limits_{i=1}^{r} n_i^2} \right) \tag{45}$$

where $i$ is the row index at $y$ and $n$, $y_i$ is the signal after being compressed using CS at row $i$, $n_i$ is the noise at row $i$, and $r$ are the number of rows from $y$.

### 3.6. Feasible Parameters

In this paper, there is more than one work to do in the signal processing environment. The first work is to encode the watermark into the secure Hadamard code. The second work is to make the host audio be a sparse signal. The third work is to hide the coded watermark into the sparse signal by CS acquisition. Thus, there are two objects for performance analysis, the detected watermark and the reconstructed audio from the detected sparse signal. From the embedded watermark relative to the length of the host audio, we can calculate the watermark payload, as described in (17). We can also calculate the CR of the sparse technique and CS performance as described in (6) from the host audio length relative to the coded and compressed audio.

Mathematically, we can simply determine the trade-off parameters between the watermark payload and the CR as presented in (17) and (6), respectively. In (17) and (6), there are the three same parameters affecting the payload and the CR, $M$, $r$, and $p$, where $M$ is the square root of the host audio length or the row/column number of the diagonal matrix ($\mathbf{S}$), $r$ is the row/column number of the truncated diagonal matrix ($\mathbf{S_r}$), and $p$ is the sample number of the compressed signal or the row number of the output of CS acquisition ($\mathbf{Y}$). First, we can see that $p$, $r$, and $M^2$ have different positions in (17) and (6). In (17), the positions of $p$ and $r$ are in the numerator, which means the decrease of $p$ and $r$ causes a lower payload. In (6), the positions of $p$ and $r$ are in the denumerator, which means the decrease of $p$ and $r$ causes a higher CR. Parameter $M^2$ also has a different position. This case certainly is a trade-off between payload and CR, for which we can find the moderate value of $p$ and $r$ to produce a high payload and high CR.

The relation between the three parameters $p$, $r$, and $M$ is such that $p \leq r < M$. Referring to (6), the above relation causes the denumerator $pr \ll 2Mr$ if $M$ has a high value; thus:

$$\text{CR} \approx \frac{M^2}{2Mr} = \frac{M}{2r}. \tag{46}$$

Note that CR for compression must be more than one; thus, $M/2r > 1$ or $r < M/2$. This means that the minimum truncation for compression is applied at half of the diagonal matrix $\mathbf{S} \in \mathbb{R}^{M\times M}$, obtaining $\mathbf{S_r} \in \mathbb{R}^{\frac{M}{2}\times\frac{M}{2}}$. Consequently, the relation of the three parameters becomes:

$$p \leq r < \frac{M}{2}. \tag{47}$$

Thus, we can exploit those three parameters in the above relation. Next, we find possible $p$ and $r$ values such that (17) reaches the maximum payload. The positions of parameters $p$ and $r$ are in the numerator of (17); thus, $r$ should be set to the maximum value or $M/2$ in order to obtain the maximum

payload and $p$ should be set to approximately $r$. Certainly, setting $r$ to the maximum value or $M/2$ obtains the minimum CR, then we have to be careful setting the $r$ parameter since it controls the trade-off between $C$ and CR. Due to its position, the $p$ parameter should be to the maximum value for reaching the maximum payload. The maximum value of $p$ is $r$. If $p = r$, then CS acquisition, as described in (4), produces an output with the same size as the input of CS. This condition is still acceptable when CR from (6) is more than one. CS acquisition still contributes to the watermarking process.

Figure 3a displays the payload versus CR with $M \in \{34, 66, 98, \cdots, 482\}$ and $r \in \{0.01M, 0.02M, \cdots, 0.5M\}$. All possibilities of the $r$ and $M$ combination with the restriction (47) are plotted as the magenta dots in Figure 3b. Blue dots in Figure 3a mean the mapping between the payload using Equation (17) and CR using Equation (6) where $p = r$, whereas magenta plus signs mean the mapping between the payload and CR where $p = 1$. The red vertical dotted line means the minimum CR or one. The green horizontal dashed line means the minimum payload or 20 bps [25]. Thus, the area with feasible payload and CR is the right side of the red vertical dotted line and the top side of the green horizontal dashed line. We see that many blue dots have a higher payload and CR than the magenta plus signs, which means the payload and CR with $p = r$ have many possibilities to reach much higher ones than the payload and CR with $p = 1$. The payload and CR mapping displayed in the blue dots where payload > 20 bps and CR > 1 in Figure 3a are obtained from $r$ and $M$ in the blue circle in Figure 3b; thus, we set $p = r$ for the experiment in the next section where the $r$ and $M$ combination values are selected from the blue circle in Figure 3b.

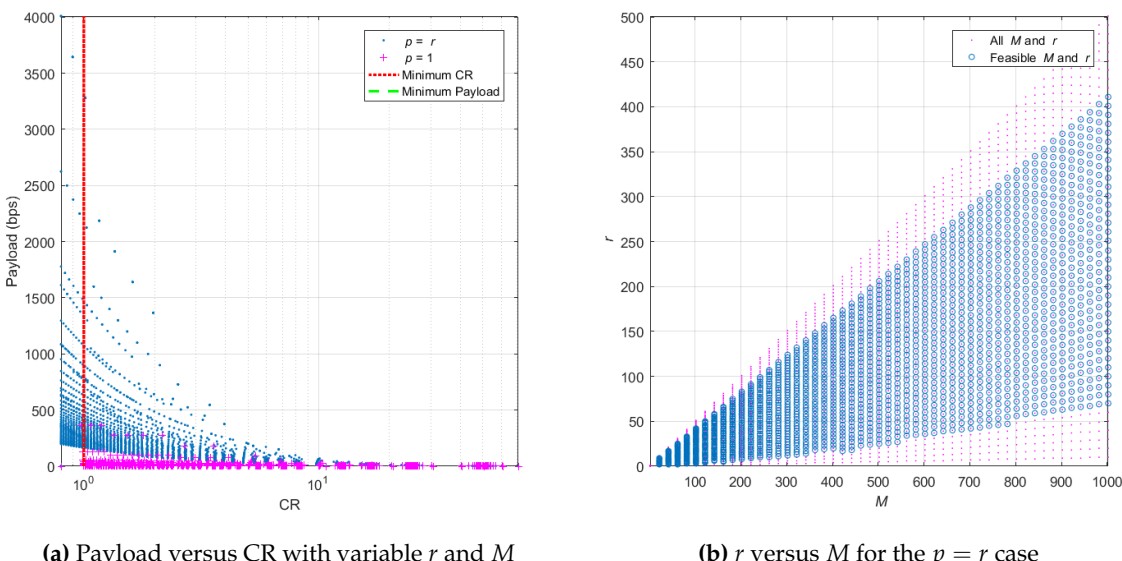

**(a)** Payload versus CR with variable $r$ and $M$      **(b)** $r$ versus $M$ for the $p = r$ case

**Figure 3.** Finding feasible $M$ and $r$ to obtain payload >20 bps and CR > 1.

## 4. Experimental Result

We assess several evaluations in this section by simulations. The evaluation aspects of the proposed method included audio quality, security, watermark quality, watermark payload, and compression ratio level aspect. The simulations ran on ASUS notebooks using MATLAB with the following specifications, Advanced Micro Devices (AMD) Fx with 12 compute cores, 16 GB Random Access Memory (RAM), and Windows 10 operating system. There were 50 mono audio host files as the clips tested with different genres of music, sampling rate 44.1 kHz and 16 bit audio quantization. All clips were in the original wave files and licensed as free audio files for research [26]. The simulation output in this section showed the average of the simulation result. The evaluated performance parameters were the audio quality, the watermark robustness, the watermark payload, and CR. The Objective Difference Grade (ODG) represents the audio quality using Perceptual Evaluation of

Audio Quality (PEAQ) [27]. Parameter *C* represents the watermark payload in bps as described in (16). Parameter BER represents the watermark robustness in (34). CR represents the Compression Ratio, as explained in (6).

We measured the audio quality between the original host audio and the reconstructed audio. The reconstructed audio quality was affected by two factors, the truncation of the diagonal matrix and the CS acquisition. The truncation of the diagonal matrix gave worse quality to the audio than the CS acquisition due to the loss of the audio signal information. The audio quality represented by ODG had a range from $-4$ to zero, where $-4$ meant the worst audio quality or the distortion was very annoying, $-3$ meant the distortion was annoying, $-2$ meant the distortion was slightly annoying, $-1$ meant the distortion was perceptible but not annoying, and 0 meant the best audio quality or the distortion was imperceptible [27].

### 4.1. Audio Quality Performance in Relation to r, M, Payload, and Compression Ratio

From Section 3.6, we selected *M* and *r* values to obtain CR > 1 and payload > 20 bps using $p = r$ as displayed in Figure 3b with the blue circle symbol. Using the selected *M* and *r* from $M \in \{34, 66, 98, \cdots, 482\}$ and $r \in \{0.01M, 0.02M, \cdots, 0.5M\}$, we applied the simulation onto five clips as the hosts. The simulation consisted of the embedding process, the data detection process, and the audio reconstruction process. It calculated the BER between the detected watermark and the original watermark, and it finally calculated the audio quality from the reconstructed audio in the ODG performance parameter. The simulation results are displayed in Figure 4a,b. From the simulation using all combinations of parameters *M* and *r* with five clips, we obtained a perfect watermark detected without any errors or BER = 0 on average. Figure 4a shows the trade-off relation between CR and payload with a negative exponential relation. Red star symbols mean the mapping between CR and payload with ODG $\geq -1$, while blue dot symbols mean the mapping between CR and payload with ODG $< -1$. We also plot the blue dots and the red stars in Figure 4b, in the relation between ODG and *M*. We can say that the longer the length of audio processed for embedding and compression, the worse the reconstructed audio quality. For the above case with five selected clips, good reconstructed audio quality or ODG $\geq -1$ was obtained when *M* < 128 samples with certain values of *r*.

The required *M* parameter did not have to be large until 482 samples, but only up to 128 samples to achieve audio quality with ODG $\geq -1$. Figure 4b shows the results. Furthermore, large *M* values had a long impact on the time processing of the insertion, detection, and reconstruction. Therefore, we applied the same simulation as the simulation displayed in Figure 4a,b using more detailed *M* and *r*, i.e., $M \in \{5, 6, ..., 128\}$, $r \in \{1, 2, ..., 64\}$, which was similar to $r \in \{0.0156M, 0.0234M, ..., 0.5M\}$ and 50 clips. We averaged the audio quality results from 50 clips, and all watermarks were perfectly detected. The simulation results are displayed in Figure 4c,d. From Figure 4d, there were many more options of *M* from five to 128, obtaining the results with ODG $\geq -1$. The simulation as displayed in Figure 4c also obtained a high CR (up to 7.03) and a high payload (up to 8296 bps). To explore which *M* and *r* obtained the above result, we also captured the simulation results in the table. Tables 4–6 respectively display the10 highest ODG, payload, and CR with certain *M* and *r*. This simulation result generally showed that we could control the audio quality, payload, and CR by adjusting the *M* and *r* parameters.

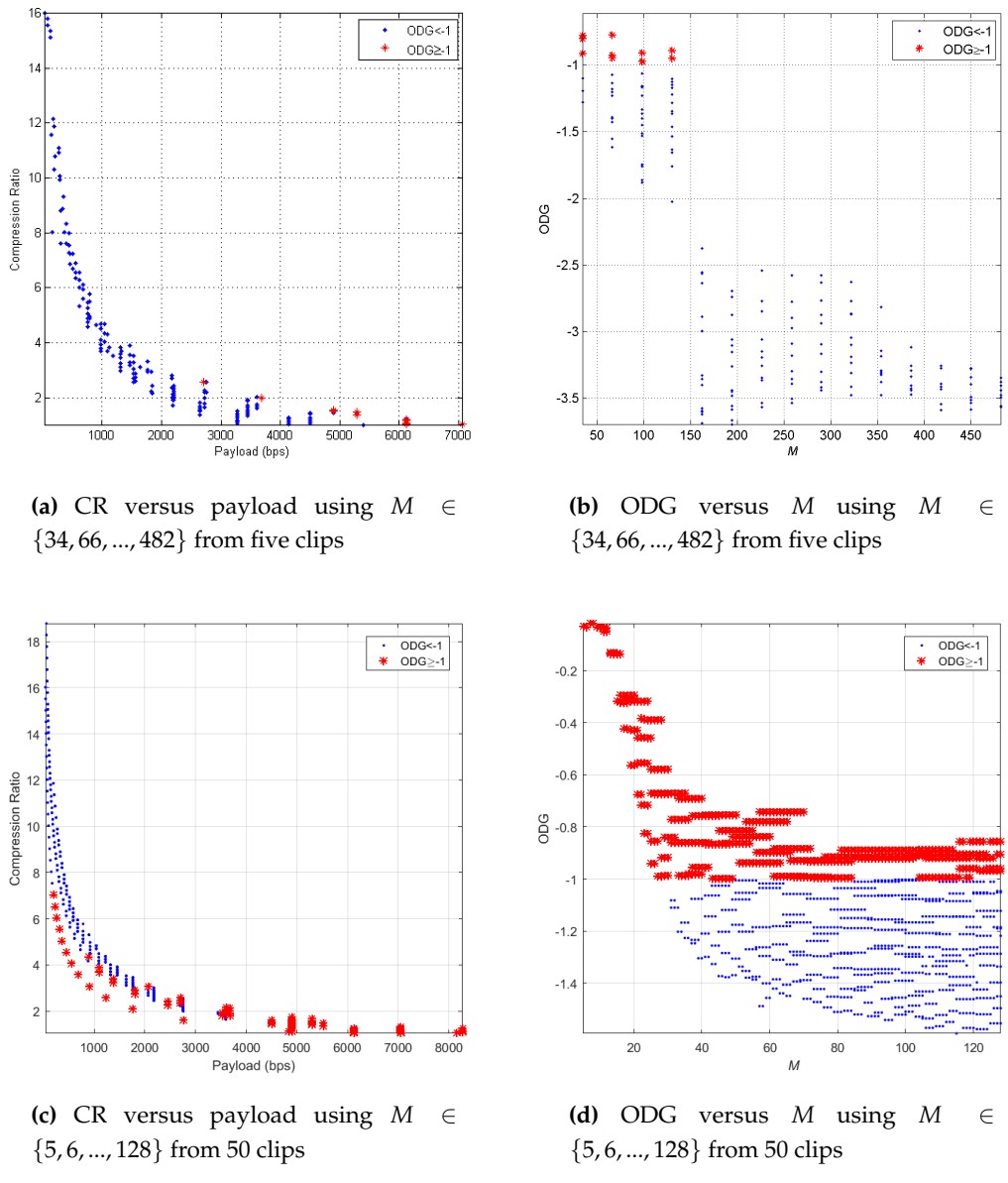

**(a)** CR versus payload using $M \in \{34, 66, ..., 482\}$ from five clips

**(b)** ODG versus $M$ using $M \in \{34, 66, ..., 482\}$ from five clips

**(c)** CR versus payload using $M \in \{5, 6, ..., 128\}$ from 50 clips

**(d)** ODG versus $M$ using $M \in \{5, 6, ..., 128\}$ from 50 clips

**Figure 4.** ODG in relation to $M$, payload, and CR.

**Table 4.** 10 highest ODG.

| $r$ | $M$ | ODG | C | CR |
|---|---|---|---|---|
| 2 | 8 | −0.02 | 2756.25 | 1.60 |
| 2 | 7 | −0.02 | 2756.25 | 1.60 |
| 2 | 5 | −0.03 | 4833.33 | 1.11 |
| 2 | 10 | −0.03 | 1764 | 2.08 |
| 2 | 9 | −0.03 | 1764 | 2.08 |
| 2 | 6 | −0.03 | 4900 | 1.12 |
| 3 | 12 | −0.04 | 5512.50 | 1.45 |
| 3 | 11 | −0.04 | 5512.50 | 1.45 |
| 3 | 10 | −0.04 | 5512.50 | 1.45 |
| 2 | 12 | −0.05 | 1225 | 2.57 |

**Table 5.** 10 highest payloads.

| r | M | ODG | C | CR |
|---|---|---|---|---|
| 5 | 20 | −0.29 | 8268.75 | 1.23 |
| 5 | 19 | −0.29 | 8268.75 | 1.23 |
| 5 | 18 | −0.29 | 8268.75 | 1.23 |
| 5 | 17 | −0.29 | 8268.75 | 1.23 |
| 5 | 16 | −0.29 | 8268.75 | 1.23 |
| 6 | 24 | −0.32 | 8268.75 | 1.14 |
| 6 | 23 | −0.32 | 8268.75 | 1.14 |
| 6 | 22 | −0.32 | 8268.75 | 1.14 |
| 6 | 21 | −0.32 | 8268.75 | 1.14 |
| 6 | 20 | −0.32 | 8268.75 | 1.14 |

**Table 6.** 10 highest compression ratios.

| r | M | ODG | C | CR |
|---|---|---|---|---|
| 2 | 30 | −0.99 | 196 | 7.03 |
| 2 | 29 | −0.99 | 196 | 7.03 |
| 2 | 27 | −0.99 | 225 | 6.53 |
| 2 | 28 | −0.99 | 225 | 6.53 |
| 2 | 25 | −0.94 | 260.95 | 6.04 |
| 2 | 26 | −0.94 | 260.95 | 6.04 |
| 2 | 23 | −0.82 | 306.25 | 5.54 |
| 2 | 24 | −0.82 | 306.25 | 5.54 |
| 2 | 22 | −0.68 | 364.47 | 5.04 |
| 2 | 21 | −0.68 | 364.46 | 5.04 |

We applied the simulation using 50 clips with $M = 32$ and $r \in \{1, 2, ..., 16\}$, which was similar to $r \in \{0.03M, 0.06M, ..., 0.5M\}$, to see how the audio truncation affected the performance parameters. Figure 5a displays the simulation result. This case also produced a perfect detected watermark or BER = 0 on average. Three performance parameters, i.e., ODG, CR, and payload as the y-axis, are displayed in one figure after being averaged, and the x-axis is the normalized rank or $r/M \in \{0.03, 0.06, ..., 0.5\}$. The black line with the right triangle symbol shows the average ODG producing $-1.16$ to $-0.16$. The blue line with a square symbol shows the payload of an embedded watermark in bps, obtaining 172.26 to 44,100 bps. The red line with a circle symbol means the CR of the encoded audio resulting from 0.20 to 7.53. The red horizontal line with the dashed-dotted symbol means the minimum CR or CR = 1. We can see that increasing the normalized rank represented by $r/M$ raised the ODG and the watermark payload, but lowered the CR of encoded audio. If the CR with the red line and circle symbol was less than the minimum CR, then it meant the CS process did not compress the audio signal overall; instead, it increased the length of the encoded signal. In this case, we could select the normalized rank less than 0.2 or $r/M \leq 0.2$ to maintain the CR to be more than one. In more detail, we could limit the minimum $r/M$ such that ODG $> -1$, i.e., $r/M \geq 0.1$. Thus, the selected range of normalized rank was $[0.1, 0.2]$, obtaining the watermark payload with the range $[729, 5292]$ bps, the compressed ratio with the range $[1.47, 4.84]$, and ODG with the range $[-0.94, -0.74]$. The $r/M$ restriction for this case maintained good quality of the reconstructed audio with a high payload and CR $> 1$.

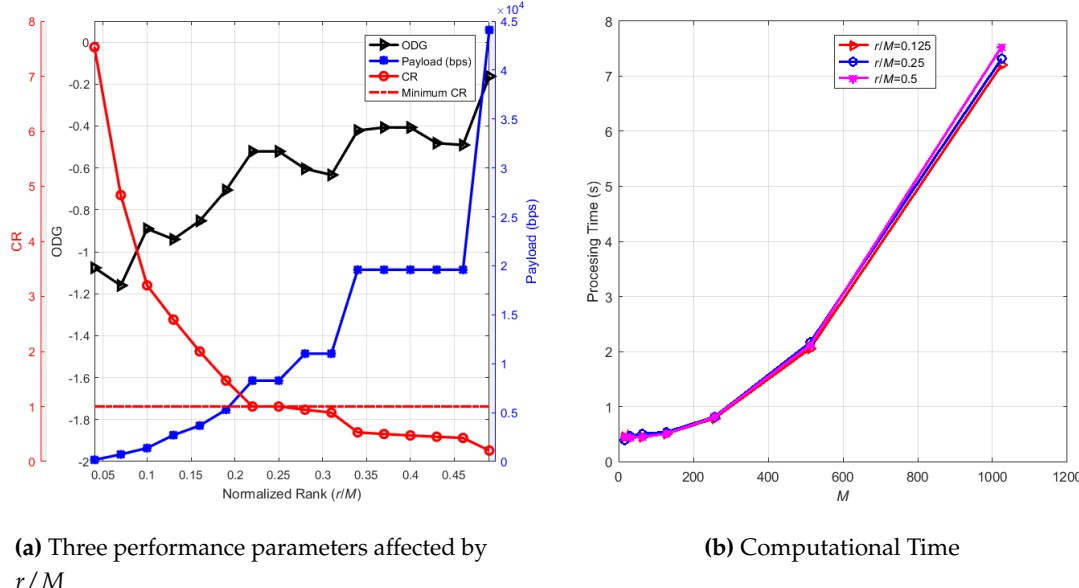

**(a)** Three performance parameters affected by $r/M$

**(b)** Computational Time

**Figure 5.** Three performance parameters affected by $r/M$ and computational time.

### 4.2. Complexity and Computational Time

The major components of the proposed data hiding and compression method in this paper consisted of DCT, the multi-bit SS mapping, singular value decomposition, and the CS acquisition process in the embedding and multi-bit SS de-mapping, SVD reconstruction, and audio decoding via CS reconstruction and IDCT. Each component had a different complexity. The SVD process to obtain $\mathbf{U} \in \mathbb{R}^{M \times M}$, $\mathbf{S} \in \mathbb{R}^{M \times M}$, and $\mathbf{V} \in \mathbb{R}^{M \times M}$ from $\mathbf{X} \in \mathbb{R}^{M \times M}$ had a complexity of $O(M^3)$ [28]. When we needed to get $\mathbf{X}$ from $\mathbf{U}$, $\mathbf{S}$, and $\mathbf{V}$ as (3), its complexity was $O(M^{2.37})$ [29]. DCT and IDCT described in (8) and (9) had a complexity of $O(N_p^2)$ where $N_p$ is the number of the DCT points, and $N_p = M$ in this case. The CS acquisition in (4), which was also the multi-bit SS embedding, had a complexity of $O(pr^2)$. The multi-bit SS detection, as described in (22), had a complexity of $O(r^3)$. Finally, the audio reconstruction by OMP approach in (40) had a complexity of $O(p^2r)$. Due to the relation $p \leq r < M$, the highest computational cost was found in the singular value decomposition, i.e., $O(M^3)$; thus, the overall complexity of the components was dominated by the SVD. This finding confirmed the reason to use the lower $M$ value. However, we still needed to check the computational time by the simulation to find out a proper $M$ value to avoid a very long processing time.

We applied the simulation to find out the computational time, which should represent the complexity of the embedding and the detection stage. In the simulation, we applied parameter $M$ from 16 to 1024 with multiples of a power of two, parameter $r = 0.125M$, $r = 0.25M$, and $r = 0.5M$. We used 10 clips in the simulation, and we averaged the time processing result. The result is displayed in Figure 5a. The processing time exponentially increased when $M$ rose. Parameter $r/M$ had no significant impact on the computational time. From this figure, lower $M$ was recommended due to the low computational time. Moreover, as confirmed in Section 4.1, the lower $M$ had a significant impact on the reconstructed audio quality.

### 4.3. Security Analysis

In Section 3.3, there were two parameters having an impact on the model security, i.e., $N_l$ as the number of the generated integer random permutation value and $r$ as the row and column number of the diagonal matrix after being truncated, $\mathbf{S_r}$. The original Hadamard matrix is denoted as $\mathbf{H_r}$, and the secured Hadamard matrix is denoted as $\mathbf{H_s}$. We applied the simulation varying $r$ and $N_l$ to understand how much $r$ and $N_l$ affected the security performance. In the real situation, one can try

to break the security model by using the original Hadamard matrix for detecting the watermark and reconstructing the audio due to the simplicity of the Hadamard matrix generation. With the secured Hadamard matrix in the encoder, we applied the decoding by the original Hadamard matrix to analyze the strength of the security model. If the security model worked well, the detection watermark should ideally be damaged, or the BER should be near 0.5.

In the simulation, we assumed $p = r = 20$ and $M = 128$ samples. $N_l$ varied from zero to $r$. Parameter $N_l$ was zero, meaning that $\mathbf{H}_s = \mathbf{H}_r$. We used five clips for analysis by calculating the average BER after the watermark detection process. We applied the simulation in 100 iterations for each clip. The simulation result is shown in Figure 6a. The worst detected watermark was obtained when $N_l$ was half of $r$, and the perfect watermark was detected when $N_l = 0$ and $N_l = r = 20$. We could limit the accepted minimum BER to restrict the value of $N_l$. We chose BER = 0.4 as a safe minimum BER because we could still interpret the digital visualization from the detected watermark with BER < 0.3 [30]. Therefore, we chose $N_l > 6$ or generally $N_l > 0.3r$ as the minimum value of $N_l$ and $N_l < 14$ or generally $N_l < 0.7r$ as the maximum value of $N_l$ to keep the detected watermark uninterpretable when one tries to detect the watermark by the original Hadamard matrix.

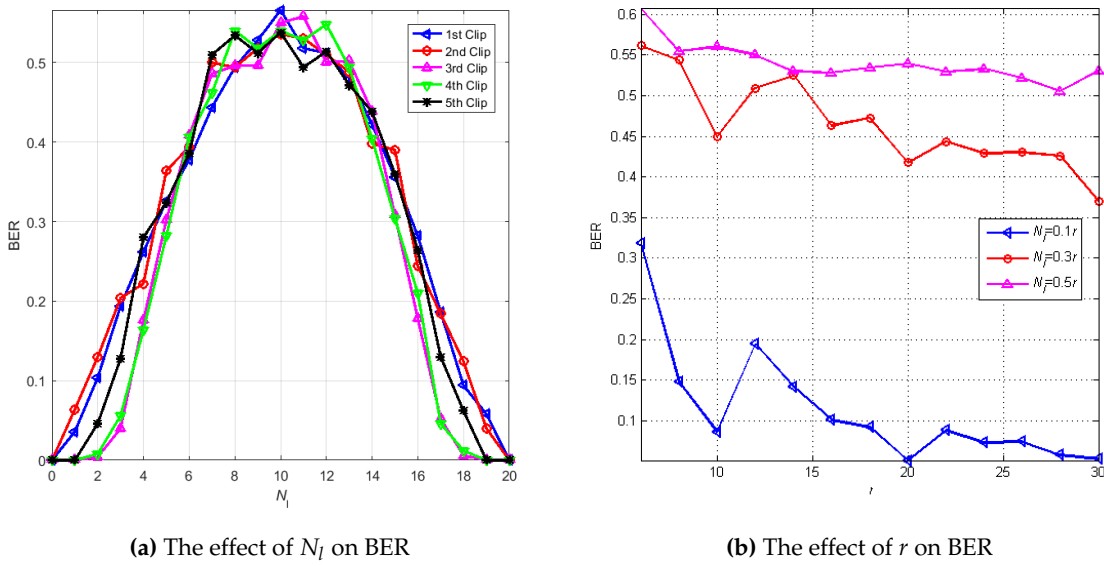

**(a)** The effect of $N_l$ on BER　　　　　　　　　　　　　　**(b)** The effect of $r$ on BER

**Figure 6.** BER in relation to $N_l$ and $r$ using a different Hadamard matrix between encoding and decoding.

Figure 6b shows the relation between BER and $r$ and compares the detected watermark quality using the different $N_l/r$. The simulation was applied to 50 clips via 10 iterations for each clip. The range of $r$ was [6, 30]. The worst watermark was detected when $N_l/r = 0.5$. The detected watermark quality was better when $N_l/r$ decreased and as the value of $r$ increased. When $N_l/r = 0.3$, most of the BER values were more than 0.4. This result confirmed the restriction of $N_l$ in the range [0.3r, 0.7r].

### 4.4. Noisy Environment

In the noisy environment, our proposed method was robust to additive noise attack as confirmed mathematically in Section 3.5. Nevertheless, it was necessary to know how robust the method was if the additive noise attacks the encoded audio by simulation. We analyzed the detected watermark quality represented by BER and the reconstructed audio quality represented by ODG as two performance parameters affected by the additive noise. In the simulation, we used 50 clips with 50 iterations for each clip, $M = 23$, $r = 6$, and $p = r$. The additive noise parameter or the input parameter for the simulation was SNR, as described in (45), whose range was 0 to 40 dB. ODG and BER as the performance parameters obtained were averaged, as displayed in Figure 7. Decreasing the noise power

or increasing the SNR rose the reconstructed audio quality or ODG and the detected watermark quality or BER.

We embedded the watermark image with the letters "ITB" and a resolution of $20 \times 35$ to understand the interpretation of the value of BER. The detected watermarks are displayed in Table 7 with various BER. We used one selected clip as the audio host using parameters $M = 256, r = 100$, and $p = r$. The original watermark image is shown at the very bottom of Table 7, since its BER was zero. We used the additive noise as the attack with various SNR from 0 to 55 dB. The detected watermark was interpretable as "ITB" when the SNR of the noise was more than 25 dB or its BER was less than 10%. Thus, the maximum acceptable BER for the detected watermark was up to 10%. In Figure 7, BER less than 10% could be achieved on an SNR of 10 dB and above. This meant that the detected watermark was already interpretable when the noise power was still half of the signal power. Furthermore, ODG was already more than $-1$. These results confirmed the robust proposed method of additive noise. The reconstructed audio was also robust to the additive noise since the ODG already achieved more than $-1$ when the SNR was still 10 dB.

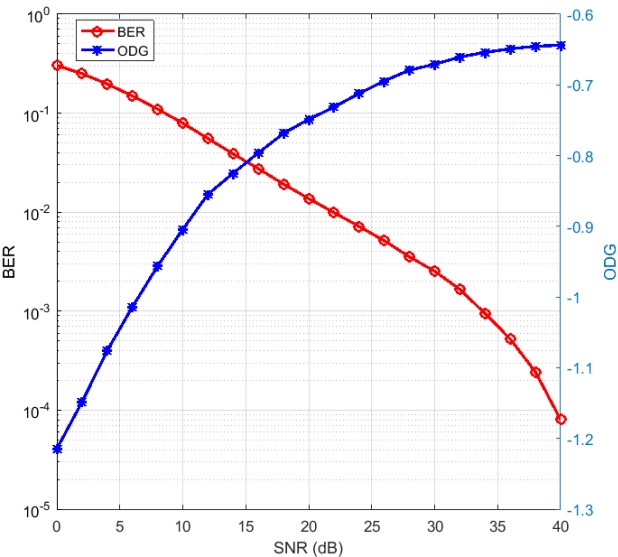

**Figure 7.** Additive noise effect and the detected watermark in certain SNR.

**Table 7.** The detected watermark in certain SNR.

| SNR | BER (%) | Detected Watermark |
| --- | --- | --- |
| 0 | 24.1 | |
| 5 | 18.7 | |
| 10 | 17.4 | |
| 15 | 14.3 | |
| 20 | 11.7 | |
| 25 | 8.8 | |
| 30 | 6.1 | |
| 35 | 5.7 | |
| 40 | 5.6 | |
| 45 | 3.6 | |
| 50 | 0.4 | |
| 55 | 0 | |

*4.5. Method Comparison to References*

As described in Section 1, there are several references related to this proposed method. We proposed a new method with more benefits than the mentioned references. Our proposed method could be used for both audio watermarking or audio steganography with compression due to the controllable parameter between the payload, the audio quality, and the compression ratio. Besides, our proposed method produced the encoded audio, which could not be attacked by a general signal processing attack, i.e., Stirmark benchmark, except the additive noise as described in Section 3.5. Table 8 displays the comprehensiveness comparison between our proposed method and the previous references, which also used CS as the embedding or compression method and the audio as the object to embed or to compress. From the previous references in Table 8, the reference [1] only described the robustness as only one performance parameter, although his method had the same purpose as our method. The reference [2] proposed the hiding method only. The reference [3] proposed the audio compression scheme only. In detail, we could only compare the performance to [4] because of its comprehensiveness performance, and the performance parameters were the closest to our method.

**Table 8.** Comprehensiveness comparison.

| Ref. | Hiding Method | Audio Reconstruction | Audio Quality | Robustness | Payload | Compression Ratio |
|------|---------------|----------------------|---------------|------------|---------|-------------------|
| [1] | Watermark Projection | × | × | √ | × | × |
| [2] | Semi-Fragile Zero Watermarking | × | × | √ | × | × |
| [3] | - | √ | √ | × | × | √ |
| [4] | Basis Pursuit Denoising | √ | √ | √ | √ | × |
| Proposed | Multi-Bit Spread Spectrum | √ | √ | √ | √ | √ |

Table 9 displays the performance comparison between our proposed method and [4]. In [4], Fakhr described the performance of four techniques. The audio quality was quite imperceptible since SNR = 28 dB. Our method was also quite imperceptible since the ODG range was [−0.94 −0.74]. Although [4] had better robustness to additive noise attack with SNR 20 dB where maximum BER = 3% and our method obtained maximum BER = 13%, our method had an outstanding payload compared to the payload in [4]. Note that the experiments in [4] only used one clip to obtain the performance in BER, SNR, and payload. In contrast, our method obtained the average performance from the simulation results of 50 clips. Our method also reported the compression ratio with the range of [1.47 4.84], while [4] did not report the compression ratio.

**Table 9.** Performance comparison.

| Ref. | Clips | Audio Quality | BER | C (bps) | CR |
|------|-------|---------------|-----|---------|-----|
| [4] | 1 | SNR = 28 dB | 0–3% | 11–344 | not reported |
| Proposed | 50 | ODG = [−0.94 −0.74] | 0–13% | 729–5292 | 1.47–4.84 |

## 5. Conclusions

In this paper, we proposed and reported a novel audio watermarking method with the CS technique, which attempted to insert the watermark into the host audio and simultaneously compressed the audio that was inserted by the watermark so that the watermarked audio had a smaller size. We also provided the security aspect of this proposed method using a secure Hadamard matrix. The proposed method worked well in a noiseless and noisy environment by mathematical derivation. Parameter performance, such as payload, CR, ODG, and BER, was reported in this paper. The experimental result showed that the proposed method presented a high imperceptibility property with payload in the range of 729–5292 bps and a compression ratio of 1.47–4.84. There was a trade-off

relation between payload and CR. We could choose the performance, specifically adapting to the requirements.

**Author Contributions:** Conceptualization, G.B. and A.B.S.; methodology, G.B., A.B.S., and D.D.; software, G.B.; validation, A.B.S. and D.D.; formal analysis, A.B.S. and D.D.; investigation, G.B.; resources, G.B.; data curation, G.B.; writing, original draft preparation, G.B.; writing, review and editing, G.B. and A.B.S.; visualization, G.B.; supervision, A.B.S. and D.D.; project administration, A.B.S. and D.D.; funding acquisition, A.B.S. All authors read and agreed to the published version of the manuscript.

**Funding:** This research was funded by Institut Technology Bandung and the Ministry of Research, Technology and Higher Education of Indonesia in 2020.

**Acknowledgments:** This research is supported by Institut Teknologi Bandung and Telkom University.

**Conflicts of Interest:** The authors declare no conflict of interest.

## Abbreviations

The following abbreviations are used in this manuscript:

| | |
|---|---|
| CS | Compressed Sensing/Compressive Sampling |
| WHT | Walsh Hadamard Transform |
| DCT | Discrete Cosine Transform |
| LSB | Least Significant Bit |
| TLC | Karhunen-Loeve Transform |
| MP3 | Motion Picture Experts Group Audio Layer 3 |
| SVD | Singular Value Decomposition |
| CR | Compression Ratio |
| IDCT | Inverse DCT |
| SS | Spread Spectrum |
| BER | Bit Error Rate |
| OMP | Orthogonal Matching Pursuit |
| SNR | Signal-to-Noise power Ratio |
| ODG | Objective Difference Grade |
| PEAQ | Perceptual Evaluation of Audio Quality |
| AMD | Advanced Micro Devices |
| RAM | Random Access Memory |

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
