# Peer review of "Compressive Sampling with Multiple Bit Spread Spectrum-Based Data Hiding"

_applsci, doi:10.3390/app10124338_

Round 1

Reviewer 1 Report

An audio watermarking method with compressive sening technique is proposed. The proposed scheme is new. The presentation is clear with fine simulations. I recommend the acceptance of this paper after minor revision.

  1. Three papers related to the compressive sensing for data hiding, the authors should introduce those papers as follows.

        1.1 Jeng-Shyang Pan, Wei Li, Chun-Shen Yang and Lijun Yan, “Image steganography based on subsampling  and compressive sensing”, Multimedia Tools and Applications, Vol. 74, No. 21, pp. 9191-9205, 2015

        1.2 Jeng-Shyang Pan, Jia-Jiao Duan, Wei Li, “A Dual Watermarking Scheme by Using Compressive Sensing and Subsampling”, ECC 2015: 381-389, 2015

        1.3 Constantinos Patsakis and Nikolaos Aroukatos, “LSB and DCT steganographic detection using compressive sensing”, Journal of Information Hiding and Multimedia Signal Processing, Vol. 5, No. 1, pp. 20-32, January 2014

2. Please also check the grammar carefully, such as "From the previous reference in Table 8, The reference [3] proposed the audio compression..." , The should be the.

Author Response

Dear Reviewer 1,

Please find the attached file as our response to the Reviewer's comments.

Thank you

Regards

Gelar Budiman

Reviewer 2 Report

The authors propose a data hiding method for audio, where simultaneuous compression
and watermark insertion is realized. Embedding is based on spread
spectrum techniques using Hadamard matrix. Emdedding method involves securing
the watermark using randomized multiplication of the Hadamard matrix. Host
signal is transferred to DCT domain and decomposed using SVD (Singular Value Decomposition)
before watermark embedding. Watermark extraction is realized from the encoded and watermarked
by detecting dictionary matrix containing the hidden data. Audio is reconstructed by OMP approach
and IDCT. The experiment cover the performance in terms of payload, compression ratio,
audio quality, and watermark quality. Audio quality, payload, and compression ratio can be controlled
with selected parameters to range which keeps the reconstructed audio in good audio quality with high payload and CR>1.
The noisy environment is considered as the only possible attack as
the compressed and watermarked audio is coded audio. Also time complexity and security is analysed
in the experiments

The article provides background, and in detail description of the algorithm. Also various
parameters and their effect on perfomance are thoroughly considered in the article. However,
comparison to references (chapter 5.5) still needs some improvement.

notes:
line 56 -> repetition of a sentence that is already in the previous chapter

4. Discussion -> Title of this section should be more descriptive
(usually section named Discussion is placed after experimental section)

5.5. Method Comparison to References
This chapter is quite superficial, needs some more details and discussion. Differences of the proposed approach
to some literature have been shortly discussed in the introduction, but here should be gone in more detail.
For example in [1], Hua proposed a data hiding technique which combined with CS synthetically. The main idea
is the same. What is the performance gain of the proposed approach

Author Response

Dear Reviewer 2,

Please find the attached file as our response to the Reviewer's comments.

Thank you

Regards

Gelar Budiman
